# The Effectiveness of the Foodbot Factory Mobile Serious Game on Increasing Nutrition Knowledge in Children

**DOI:** 10.3390/nu12113413

**Published:** 2020-11-06

**Authors:** Hannah M. Froome, Carly Townson, Sheila Rhodes, Beatriz Franco-Arellano, Ann LeSage, Rob Savaglio, Jacqueline Marie Brown, Janette Hughes, Bill Kapralos, JoAnne Arcand

**Affiliations:** 1Faculty of Health Sciences, Ontario Tech University, 2000 Simcoe Street North, Oshawa, ON L1G 0C5, Canada; Hannah.froome@ontariotechu.net (H.M.F.); Carlissa.townson@ontariotechu.net (C.T.); Beatriz.FrancoArellano@ontariotechu.ca (B.F.-A.); Jacquelinem.brown@sickkids.ca (J.M.B.); 2Faculty of Education, Ontario Tech University, 2000 Simcoe Street North, Oshawa, ON L1G 0C5, Canada; Sheila.Rhodes@ontariotechu.ca (S.R.); Ann.lesage@ontariotechu.ca (A.L.); Janette.hughes@ontariotechu.ca (J.H.); 3Faculty of Business and Information Technology, Ontario Tech University, 2000 Simcoe Street North, Oshawa, ON L1G 0C5, Canada; Robert.savaglio@ontariotechu.net (R.S.); Bill.Kapralos@ontariotechu.ca (B.K.)

**Keywords:** healthy eating, digital interventions, serious games, nutrition education, nutrition knowledge, Canadian children

## Abstract

The interactive and engaging nature of serious games (i.e., video games designed for educational purposes) enables deeper learning and facilitates behavior change; however, most do not specifically support the dissemination of national dietary guidelines, and there are limited data on their impact on child nutrition knowledge. The Foodbot Factory serious game mobile application was developed to support school children in learning about Canada’s Food Guide; however, its impacts on nutrition knowledge have not been evaluated. The objective of this study was to determine if Foodbot Factory effectively improves children’s knowledge of Canada’s Food Guide, compared to a control group (control app). This study was a single-blinded, parallel, randomized controlled pilot study conducted among children ages 8–10 years attending Ontario Tech University day camps. Compared to the control group (*n* = 34), children who used Foodbot Factory (*n* = 39) had significant increases in overall nutrition knowledge (10.3 ± 2.9 to 13.5 ± 3.8 versus 10.2 ± 3.1 to 10.4 ± 3.2, *p* < 0.001), and in Vegetables and Fruits (*p* < 0.001), Protein Foods (*p* < 0.001), and Whole Grain Foods (*p* = 0.040) sub-scores. No significant difference in knowledge was observed in the Drinks sub-score. Foodbot Factory has the potential to be an effective educational tool to support children in learning about nutrition.

## 1. Introduction

High nutritional quality diets are fundamental to optimal child physical growth and cognitive development [1,2]. Diets of lower nutritional quality are associated with lower cognitive and academic achievement, and a higher risk of developing long-term health issues such as type 2 diabetes, hypertension, and heart disease [3,4]. Moreover, the prevalence of overweight and obesity has risen among Canadian children and adolescents [5], which itself is associated with lower academic performance, feelings of social isolation, anxiety and depression, and low self-esteem and body image issues [6,7,8]. Many Canadian children consume excess sodium and sugar from packaged foods and beverages and insufficient amounts of beneficial foods and nutrients including protein, whole grain foods and vegetables [9], which are dietary habits that may carry over from childhood to adulthood [10]. Subsequently, health promotion interventions to support the acquisition of nutrition knowledge, to improve nutrition literacy, and develop food skills during childhood are warranted in order to prevent negative health outcomes associated with poor quality diets. Nutrition knowledge and skills may be further enhanced by school-based nutrition policies and curriculum-based nutrition education [11], which are strategies employed in most jurisdictions within Canada; however, few studies have captured the impacts of policies [12].

The use of gameplay through serious games to educate about nutrition is an emerging area in the nutrition education literature. Recent studies have shown that engaging apps, and the use of gamification and behavioural change techniques may increase intake of vegetables and fruits and increase engagement in food-related conversations in children [13,14,15,16,17]. In particular, serious games (i.e., video games designed for educational purposes) [18] have been shown to encourage participation and motivation in learning activities, which can promote higher academic achievement among elementary school children [19]. However, there are few evidence-based serious games that exist to support nutrition education for children and none known to support health promotion efforts in disseminating dietary guidelines [20,21]. In 2019, Health Canada published a new Canada’s Food Guide (CFG), which was disseminated in an online-first format, with the majority of the CFG content published on a dedicated CFG website as opposed to traditional paper-based leaflets [22,23]. While this approach has the potential to facilitate access to nutrition information, the CFG website is didactic in nature and lacks engaging and motivating features, such as gamification and behavior change techniques (e.g., avatars, games, quizzes, and rewards), known to enable deeper learning among children [24].

To address these gaps, the Foodbot Factory mobile application was developed by a team of experts in nutrition sciences, dietetics, game development/computer science, and educational pedagogy and technology at Ontario Tech University in Oshawa, Canada [22]. Foodbot Factory is an evidence-based serious game that aims to engage children ages 8–10 years in learning about healthy eating and the new CFG, and has been available in the Google Play store since June 2020 [25]. Foodbot Factory contains five learning modules related to four CFG food groupings: Drinks, Whole Grain Foods, Vegetables and Fruits, and Protein Foods; the latter is organized into two learning modules: Animal Protein Foods and Plant-based Protein Foods [22]. Each learning module has integrated game elements and behavior change techniques that are used to enhance engagement with the app content (i.e., games within a game, humorous dialogue, a food log, quizzes) [22]. Foodbot Factory has shown positive results in facilitating learning as part of the iterative development process [22]; however, no empirical evaluation has been conducted to determine whether it improves nutrition knowledge. Therefore, the objective of this study was to assess whether Foodbot Factory increases overall nutrition knowledge in children compared to a control app. Secondary outcome measures included changes in nutrition knowledge in each of the four sub-scores of Vegetables and Fruits, Whole Grain Foods, Protein Foods, and Drinks. A detailed assessment of learning was also conducted when children used Foodbot Factory, which was collected to provide data to inform future iterations of the app.

## 2. Materials and Methods

### 2.1. Study Design and Setting

This study was a single-blinded, parallel, randomized controlled pilot study conducted during July and August 2019 (Ontario Tech University Research Ethics Board File #15385). It was conducted in a single centre, at the Ontario Tech University summer day camps, which consisted of Coding, Tech-based, and Minecraft camps.

### 2.2. Inclusion and Exclusion Criteria and Recruitment

Children were eligible to participate in the study if they were 8–10 years old and were entering grades 4 or 5 in September 2019. Only children who could read and write English were included. Participants were recruited from Ontario Tech University day camps. An invitation to participate in the study was sent to the parents of children who were enrolled in the camps. Informed consent was provided by parents, and assent was provided by children prior to participation.

### 2.3. Study Interventions and Protocol

Children were randomized to play Foodbot Factory or a control app called “My Salad Shop Bar” [26] for 10–15 min each day over a five day period (Figure 1). A different Foodbot Factory module and control app level was planned for each day of the study, facilitated by trained study personnel who followed a standardized protocol. The protocol was developed in collaboration with teachers and designed to mimic a classroom setting. Children in each group used the serious games on Android tablets that were provided by study personnel.

### 2.4. Randomization and Allocation

Participants were randomized into the intervention or control group at study baseline at a 1:1 allocation ratio. Randomization was stratified based on gender and grade entry in September 2019, as these were considered potential confounding variables of the primary endpoint. Randomization was conducted by the research coordinator using a computerized random letter sequence generator, with sequences in blocks of four letters.

Intervention Group (Foodbot Factory): A different learning module of Foodbot Factory was played on each day of the study: Drinks on study Day 1; Whole Grain Foods on study Day 2; Vegetables and Fruits on study Day 3; Animal Protein Foods on study Day 4; and Plant-based Protein Foods on study Day 5 (Figure 2). Each module was played for 15 minutes. A voiceover was available to Foodbot Factory in order to enhance overall engagement and make the content more accessible for those who may have difficulty reading. The iterative development process for Foodbot Factory has been described elsewhere [22].

Control Group: The control group played the mobile application “My Salad Shop Bar” [26]. This is a food-focused cooking game where the player prepares an order of healthy food (i.e., salads, fruit smoothies, whole grain breads, etc.) for customers. This app was chosen because it is a gamified mobile app that exposes the participants to a wide range of healthy foods. Each level takes up to five minutes to play; however, children were provided the opportunity to play the app for 15 minutes to ensure consistency with the intervention group.

For both the intervention and control groups, the protocol design took additional steps to minimize any factors that may introduce bias, confounding, co-intervention, and contamination. At the time the study was conducted, the Foodbot Factory app was not publicly available, so there was no risk for co-intervention. Participants in both study groups used the applications for the same duration of time in sessions that were led by a facilitator. Participants and parents were blinded to which app was the intervention and which was the control. To reduce the risk of contamination, the children in the control and intervention groups were separated into different rooms during the study period. Possible confounding factors such as mobile device use at home and the use of nutrition apps were measured, and any noticeable changes in the children’s knowledge and behaviour at home was also measured through a questionnaire sent to parents. A research assistant was located within the intervention and control room at all times and was able to observe/monitor for participant adherence for the duration of the study.

### 2.5. Primary and Secondary Outcomes

The primary outcome was change in overall nutrition knowledge, which was assessed at study baseline (Day 1) and at the end of the study period (Day 5). The secondary outcomes were change in nutrition knowledge for each of the four sub-scores as measured with the validated Nutrition Attitudes and Knowledge (NAK) Questionnaire, which was designed for and validated with primary school children. The NAK questionnaire was designed to capture changes in nutrition knowledge from use of the Foodbot Factory and took approximately 10–15 minutes to complete. The NAK is organized into four sub-scores that align with the CFG food groupings: Drinks, Whole Grain Foods, Vegetables and Fruits, and Protein Foods (Animal and Plant-based protein foods combined). The NAK questionnaire is comprised of 20 questions overall, with five questions allocated to each of the four sub-scores. The maximum score was 20 for overall knowledge. The maximum score was 5 for each of the sub-scores. All questions are equally weighted. Questions are multiple choice and true/false. The sum score for each of the four sub-scores provides an overall nutrition knowledge score. Prior to randomization, parents completed a questionnaire that collected children’s baseline demographic information and mobile device use/habits. The child’s body mass index (BMI) was calculated based on parent-reported weight and height. At the completion of the study, parents completed an end-of-study questionnaire that captured any co-interventions or outside influences that may have affected the children’s learning, and whether the children had any changes in their interest in food and nutrition at home (Figure 1).

### 2.6. Sample Size Calculation

The sample size calculation was based on data from a proof-of-concept study among children of a similar age who used the Foodbot Factory app. Considering an effect size of 3.2, a standard deviation of 3.2 and 80% power, 34 participants per group were required. The criterion for statistical significance was set at <0.05. We assumed a 20% attrition rate and recruited additional subjects to account for this consideration. This also accounted for any participants who were removed from the analysis.

### 2.7. Statistical Analyses

Continuous data are presented as means and standard deviations, and categorical data are presented as frequencies and percentages. Baseline characteristics between the two groups were analyzed using unpaired t-tests (continuous data) and the Chi-square test (categorical data). For continuous data, including the primary endpoint and secondary endpoints, two-way analysis of variance was measured between and within group differences in nutrition knowledge before and after the study period. A post hoc Tukey test was used to assess for the differences in time and treatment. A Chi-square test was conducted on the assessment of the correct answers in participants who used Foodbot Factory. A *p*-value of <0.05 was considered statistically significant. Data analyses were conducted in using the statistical software IBM SPSS Statistics 26. Intention-to-treat analysis was not conducted as there was a small sample of participants who dropped out or were excluded from the study.

## 3. Results

### 3.1. Participants

A total of 496 of the children enrolled in the Ontario Tech University day camps were assessed for eligibility. Emails were then sent to the parents of 310 eligible children who met the study criteria, inviting their child to participate in the study. In total, 95 participants (34% recruitment rate) with consenting parents were enrolled and randomized in the trial. Overall, 22 participants were removed from the study: One participant from both the Foodbot Factory and control group dropped out, and ten participants from both the Foodbot Factory and control group were excluded from the analysis due to being absent from the camp for at least 2 days (*n* = 15), or being absent when the study outcomes were assessed (*n* = 5). The final data sample included 73 participants (Figure 3).

### 3.2. Baseline Characteristics

Participants’ baseline characteristics are described in Table 1. Overall, a higher proportion of participants (62%) were boys with an average age of 9.0 ± 0.8 years, with the prevalence of obesity being higher in the control group (13% vs. 21%, *p* = 0.031). Significantly more participants in the control group (41% vs. 18%, *p* = 0.029) used a mobile device for learning at school. Most parents claimed that they often encouraged their children to follow a healthy diet (Foodbot Factory; 80%, Control; 79%, *p* = 0.994) and often talked about food and nutrition with their child (Foodbot Factory; 54%, Control; 74%, *p* = 0.084).

### 3.3. Changes in Overall Nutrition Knowledge and Sub-Scores of Knowledge

There were no baseline differences in overall nutrition knowledge or in any of the sub-scores between children randomized to Foodbot Factory and the control group. Baseline nutrition knowledge scores were relatively low in both the Foodbot Factory and control group. During the intervention period, a statistically significant increase in overall nutrition knowledge was observed in the Foodbot Factory group (10.3 ± 2.9 to 13.5 ± 3.8), compared to the control group (10.2 ± 3.1 to 10.4 ± 3.2, *p* < 0.001, Table 2, Figure 4a). Significant increases in nutrition knowledge were also observed in the Foodbot Factory group, compared to the control group, related to sub-scores of Vegetables and Fruits (*p* < 0.001), Protein Foods (*p* < 0.001), and Whole Grain Foods (*p* = 0.040). No significant increase in knowledge was observed between the two groups in the Drinks (*p* = 0.206); however, baseline knowledge of Drinks was already relatively high in both the Foodbot Factory (3.8 ± 1.0) and control (3.5 ± 1.2) groups (Table 2, Figure 4b–e).

### 3.4. Changes within the Foodbot Factory Application Questions

Further analyses were conducted to understand where exactly changes in nutrition knowledge occurred when children used Foodbot Factory. Table 3 shows a detailed assessment of correct answers among participants who used Foodbot Factory. For example, there was more than a 100% increase in correct responses to questions related to different types of fats and heart disease risk, types of fats present in animal protein foods, the proportion of a plate/meal that should be vegetables and fruits, and limiting canned vegetables and fruits that are high in sodium and/or sugar. While there were widespread improvements in nutrition knowledge after using Foodbot Factory, there were also individual items (questions) within each sub-group where little improvement was observed such as the nutrients found in dairy and soy milk, why drinking water is important, and what types of whole grain foods to choose most often. These individual items reflect more specific knowledge about a concept within a sub-group. These data can be used to inform future iterations of the app.

### 3.5. Changes in Nutrition Interests during the Study Period

Changes in nutrition-related interests that occurred during the study period were assessed, and any co-interventions that may have occurred, via a parent questionnaire at the end of the study. No significant between-group differences were observed (Table 4). However, 62% of parents in the control group and 41% of parents in the Foodbot Factory group reported that their child mentioned food, diet, or nutrition “more than usual” in conversation.

## 4. Discussion

This study has shown that Foodbot Factory, a novel evidence-based serious game mobile application designed to teach children about Canada’s Food Guide, has the potential to be an effective learning tool that can result in long-term knowledge and nutrition outcomes in school-aged children. The need for novel digital tools for health promotion has been documented by several sources [27,28] and is clearly substantiated in light of the immediate transition to digital learning during the COVID-19 pandemic [27,29]. The children who used Foodbot Factory over a five-day period for an average of 15 minutes of nutrition education per day had significantly greater overall nutrition knowledge and sub-scores of nutrition knowledge (Vegetables and Fruits, Protein Foods, and Whole Grain Foods), compared to children who used the control application. At the end of the study period, participants received, in total, 75 minutes of nutrition education.

Improving food skills and nutrition knowledge in children are among several approaches to enable healthy eating behaviours during childhood and beyond [10,30,31]. Nutrition knowledge and food skills enable individuals (youth and adults) to navigate complex food environments through evaluating and interpreting nutrition information to make healthy eating choices [31]. Nutrition education aims to encourage student confidence in their abilities to make healthy eating decisions, build skills to enhance their dietary behaviours, and understand factors, such as social media, that affect their food choices [11,32]. Despite the inclusion of nutrition as part of the Canadian school curriculum [32], an understanding of children’s use of nutrition information to assess food environments and make food decisions among children is minimal, and there is limited literature on what exactly Canadian children know about nutrition. Improving nutrition knowledge is a crucial public health priority and well-designed evidence-based digital interventions, especially those that involve gameplay via serious games, are an increasingly common and a highly effective method to engage children in learning, making education fun and enjoyable.

Research on serious games in health promotion is an emerging field, with studies demonstrating their impact on nutrition knowledge and behaviours [14,33,34]. Many of the studies assessed reported positive impacts on nutrition knowledge, and health behaviour habits such as fruit and vegetable consumption, and/or intake of processed snacks [15,35,36,37]. These studies align with findings conducted by systematic reviews [13,14,38,39], which have demonstrated that digital tools were efficacious in increasing nutritional knowledge, consumption of fruits and vegetables, nutrients, and decreasing consumption of sugary drinks. Whether the latter were multi-component or stand-alone interventions, all digital tools contained at least one form of gamification or behaviour change elements such as rewards, interactive elements and skill based games to motivate and establish healthy eating behaviours [14,36,37,39]. In Canada, research on the impact of serious games on nutrition knowledge in children within the school environment is limited. Of 155 studies on serious games conducted through a systematic review, only 22 of these studies included digital and eHealth school-based interventions that assessed consumption of fruits and vegetables (*n* = 9), and consumption of sugar-sweetened beverages or snacks (*n* = 4) [14].

Currently, Foodbot Factory is the first novel, comprehensive serious game, based on the CFG, which has potential for improving nutrition knowledge amongst school-aged children in an engaging and intrinsically motivating way. As a serious game, Foodbot Factory incorporates various gamification elements and behavioural change techniques including feedback and monitoring, social support, shaping knowledge, natural consequences, reward, and threat, quizzes and sub-games requiring a user to catch food and sort food [22]. These features are known to promote motivational learning, improve knowledge and healthy eating behaviour change in children, and promote engagement, which can increase educational achievement, and is likely responsible for the positive findings observed in the present study [18,29,39,40]. The success of well-designed serious games is supported by self-determination theory, in which intrinsic motivation (engagement due to interest) is separated from extrinsic motivation (based on rewards, avoiding punishments, etc.) [39,41,42]. Serious games also address psychological needs to support motivation and well-being, such as promoting autonomy, competence, and relatedness [41,42]. Considering the existing literature and positive findings observed in the present study, the data overall support the use of serious games in motivating children to learn about nutrition and healthy eating. Importantly, there is an opportunity to consider serious games as a health promotion tool, as opposed to didactic websites that currently exist for the CFG. Dissemination as a public health policy intervention, and the broader implementation of this tool used by educators, parents, and caregivers, can support the digital implementation of national dietary guidelines and addresses the lack of research the impact of digital tools has on improving nutrition knowledge.

The importance of technology in education has been emphasized in light of school closures globally amid the COVID-19 pandemic, with children learning at home via online classrooms [43]. The need for and use of technology make health-focused serious games and eHealth tools a highly strategic avenue to address health, nutrition, and physical education. Evidence-based serious games, such as Foodbot Factory, that are comprehensive in scope have the potential to effectively promote deep nutrition-related learning. Yet, how these tools are implemented in schools requires further research. Although Foodbot Factory was tested in schools during the iterative development processes, implementation science approaches to empirically and qualitatively evaluate the real-life adoption and use of Foodbot Factory (and similar types of digital tools) are warranted in these settings. Adopting innovative game-based approaches as nutrition education interventions (and informing future iterations of such tools including augmented, virtual, or mixed realities) hold the potential to effectively increase nutrition knowledge by educating children about healthy eating [39].

This study has some limitations that need to be raised. In particular, this study was not designed to capture changes in dietary behaviours or health; however, our detailed assessments in knowledge gained provide a basis for conducting future research on these outcomes. This study was also short in duration, and not conducted in a classroom setting with a teacher and accompanying lesson. However, this study protocol was developed to mimic the classroom setting with regards to facilitation, the technology used (tablets), and the length of time participants played both apps were kept consistent. Exposure to nutrition apps could have impacted the findings. In particular, we found that significantly more children in the Foodbot Factory group used mobile apps, compared to those in the control group (74% vs. 41% *p* = 0.002). We did not assess the type of nutrition apps used (i.e., food-themed games, trackers, recipe apps). Importantly, there was no difference in nutrition knowledge between the two study groups at baseline, nor any parent-reported co-interventions over the study period. Furthermore, the study sample may not be representative of the general population of Canadian children due to the small sample size and being conducted at a tech-based day camp. While we controlled for gender during randomization, the sample included 62% and 63% of boys in the intervention and control group, respectively. Further analysis with a longer duration and larger sample size needs to be conducted to fully evaluate the long-term effectiveness of Foodbot Factory on nutrition knowledge and outcomes in children. Additionally, whether the participants were aware and had learned about the 2019 CFG prior to the study was not assessed; however, baseline knowledge was similar between the intervention and control group.

This protocol was developed in collaboration with teachers and reflected the classroom experience as much as possible. In fact, the results of this study may have underestimated the educational impact of Foodbot Factory in a classroom setting, as the app would be accompanied with teacher instruction and related activities and assignments. A major strength of this study is that it was the first known randomized study in Canada to evaluate the effectiveness of a digital engaging evidence-based serious game on improving nutrition knowledge and the new content of the 2019 CFG.

## 5. Conclusions

This study has shown that Foodbot Factory has the potential to be an effective digital tool to engage children in learning about nutrition, resulting in significant improvements in nutrition knowledge overall and across most sub-scores of nutrition knowledge. These data demonstrate the broader use of serious games to support the dissemination of healthy eating dietary guidance and in facilitating learning about healthy eating among children in elementary school and home environments. The need for novel digital tools in both global public health and health promotion interventions has been documented [27,28]. Future research needs to include assessments of changes in long-term healthy behaviours and dietary intake in response to using digital applications to fully understand the effectiveness of this digital learning tool. Data generated in this study can support evidence on the development of future digital tools and the use of gamification and behavioural change techniques to positively influence children’s nutrition education worldwide.

## Figures and Tables

**Figure 1 nutrients-12-03413-f001:**
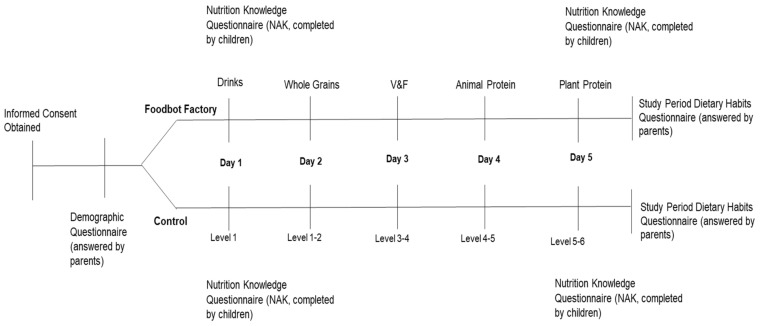
Study protocol for the week-long randomized controlled trial.

**Figure 2 nutrients-12-03413-f002:**
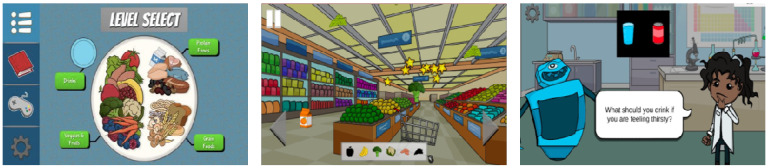
Screenshots of Foodbot Factory.

**Figure 3 nutrients-12-03413-f003:**
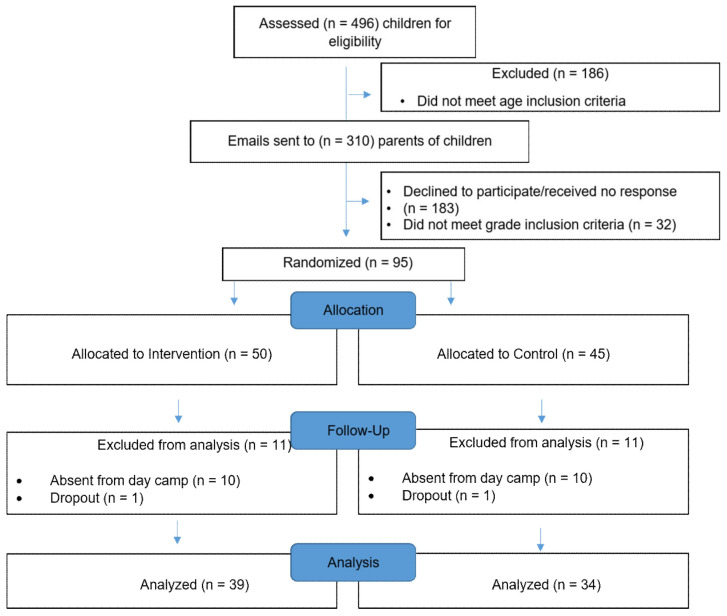
CONSORT Flow Diagram of subjects participating in the randomized controlled trial.

**Figure 4 nutrients-12-03413-f004:**
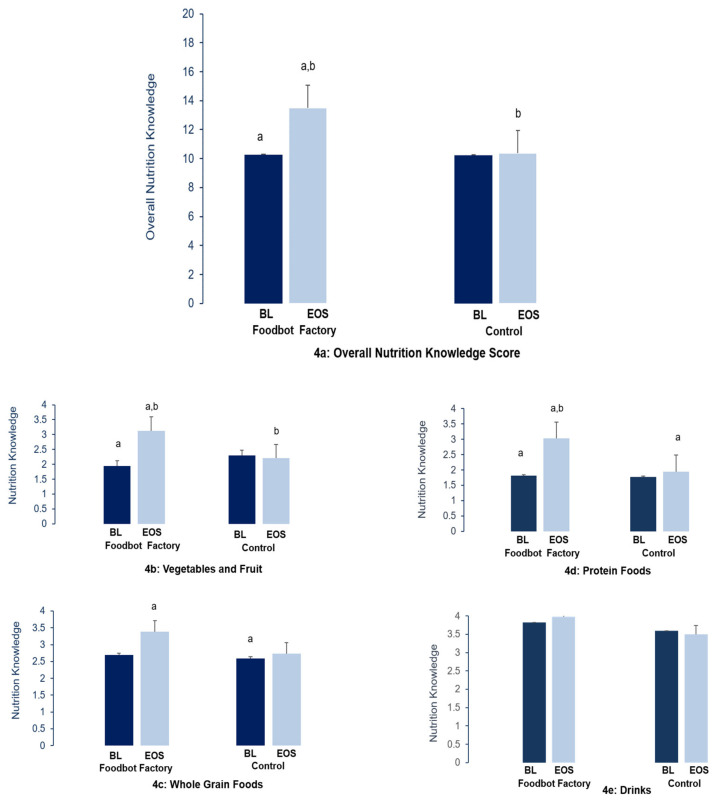
(**a**) Presents data overall Nutrition Knowledge Score. (**b**–**e**) Presents data on the sub-scores for vegetables and fruit, whole grains, protein foods, and drinks. EOS: End-of-Study. BL: Baseline. ^a,b^ Between and within group differences lie.

**Table 1 nutrients-12-03413-t001:** Baseline Demographics and Mobile Device Use.

	Foodbot Factory (*n* = 39)	Control (*n* = 34)	*p*-Value ^a^
Age (years)	9.1 ± 0.7	8.9 ± 0.7	0.474
Grade (School)	4.4 ± 0.5	4.4 ± 0.5	0.643
Proportion in Grade 4	22 (55)	21 (62)	0.643
Boys	25 (63)	21 (62)	0.836
Male sex at birth	24 (60)	21 (62)	0.984
Body mass index (BMI, kg/m2)	18.1 ± 4.6	19.9 ± 6.7	0.661
BMI Percentile Categories			
Underweight: <5th percentile	4 (10)	1 (3)	0.045
Normal Weight: 5th to <85th percentile	23 (59)	13 (33)	0.047
Overweight: 85th to ≤95th percentile	5 (13)	5 (15)	0.062
Obese: >95th percentile	5 (13)	7 (21)	0.031
Child has access to a smartphone or tablet at home	37 (95)	32 (94)	0.890
Ways in which child accesses a smartphone or tablet			
They use a parent/guardian’s device	13 (33)	14 (41)	0.489
They use a device at school	7 (18)	14 (41)	0.029
They use another adult’s device	2 (5)	2 (6)	0.888
Child owns a device of their own	25 (64)	23 (67)	0.754
Not applicable	2 (5)	2 (6)	0.888
Frequency of use of nutrition mobile apps			
4 or more times a week	1 (3)	0 (0)	0.347
3 times a week	0 (0)	2 (6)	0.125
2 times a week	0 (0)	1 (3)	0.281
Less than once/week	5 (13)	12 (35)	0.023
None	17 (44)	7 (21)	0.037
Not applicable	16 (41)	12 (35)	0.615

Categorical data are presented as frequency (percentage). Continuous data are presented as mean SD. ^a^
*p*-values were calculated with Chi-square (categorical data) and unpaired *t*-tests (Continuous data).

**Table 2 nutrients-12-03413-t002:** Changes in overall nutrition knowledge and sub-scores of knowledge.

	Maximum Score	Foodbot Factory (*n* = 39)		Control (*n* = 34)		*p*-Value ^c^
		BL	EOS	BL	EOS	
Overall Nutrition Knowledge Score	20	10.3 ± 2.9 ^a^	13.5 ± 3.8 ^a,b^	10.2 ± 3.1	10.3 ± 3.2 ^b^	<0.001
Whole Grain Foods Sub-score	5	2.6 ± 1.3	3.3 ± 1.4	2.5 ± 1.2	2.7 ± 1.1	0.040
Vegetables and Fruits Sub-score	5	1.9 ± 1.0 ^a^	3.1 ± 1.5 ^a,b^	2.2 ± 1.3	2.2 ± 1.1 ^b^	<0.001
Protein Foods Sub-score	5	1.8 ± 1.0 ^a^	3.0 ± 1.6 ^a,b^	1.7 ± 1.0	1.9 ± 1.1 ^b^	<0.001
Drinks Sub-score	5	3.8 ± 1.0	3.9 ± 0.74	3.5 ± 1.2	3.5 ± 1.1	0.206

BL = Baseline; EOS = End-of-Study. ^a,b^ indicates statistically significant within ^a^ and between ^b^ group differences. ^c^ As determined by the two-way analysis of variance.

**Table 3 nutrients-12-03413-t003:** Detailed assessment of correct answer among participants who used Foodbot Factory (*n* = 39).

	Baseline	End of Study	% Difference	*p*-Value ^a^
Drinks				
Q1: Best choice to drink when thirsty	36 (92)	39 (100)	8%	0.077
Q2: Drink that should be enjoyed less often	37 (95)	37 (95)	0	1.000
Q3: Nutrient found in dairy & soy milk	13 (33)	10 (26)	−23%	0.411
Q4: Why drinking water is important	30 (77)	31 (80)	3%	0.784
Q5: Fruit juices are a sugary drink (T/F)	33 (85)	38 (97)	15%	0.048
Whole Grain Foods				
Q6: Grain foods to choose most often	30 (77)	32 (82)	7%	0.913
Q7: Examples of whole grains include	15 (39)	24 (62)	60%	0.039
Q8: Nutrients in whole grain bread	26 (67)	34 (87)	31%	0.016
Q9: Why fibre is important for health	13 (33)	18 (46)	38%	0.174
Q10: Nutritional content of refined grains	21 (54)	24 (62)	14%	0.576
Vegetables and Fruits (V&F)				
Q11: V&F are a good source of	19 (49)	22 (56)	16%	0.569
Q12: Consume V&F that are different	23 (59)	30 (77)	30%	0.110
Q13: Nutrient found in fruits vs. fruit juice	10 (26)	16 (41)	60%	0.347
Q14: Proportion of V&F on your plate/for a meal	12 (31)	27 (69)	125%	0.003
Q15: Limit canned V&F containing added	12 (31)	27 (69)	125%	0.003
Protein Foods				
Q16: Fat of type and heart disease risk	10 (26)	26 (67)	160%	0.001
Q17: Protein food to choose most often	22 (56)	30 (77)	36%	0.035
Q18: Protein foods that are a source of fibre	17 (44)	18 (46)	6%	0.578
Q19: Nutrients in processed meats	15 (39)	24 (62)	60%	0.030
Q20: Animal protein is source if unsaturated fat (T/F)	7 (18)	20 (51)	185%	0.001

Data presented as frequency (percentage), Q = Question. ^a^ As determined by the Chi-square test.

**Table 4 nutrients-12-03413-t004:** Changes in nutrition interests during the study period.

	Foodbot Factory (*n* = 39)	Control (*n* = 34)	*p*-Value ^a^
Played food/nutrition apps or games at home			
Less than usual	2 (5)	1 (3)	0.639
No more than usual	28 (72)	23 (68)	0.700
More than usual	0 (0)	2 (6)	0.125
Don’t know	5 (13)	1 (3)	0.125
Not applicable	4 (10)	7 (21)	0.218
Mentioned food, diet, or nutrition in conversation			
Less than usual	0 (0)	0 (0)	
No more than usual	19 (49)	6 (18)	0.005
More than usual	16 (41)	21 (62)	0.077
Don’t know	0 (0)	0 (0)	
Not applicable	4 (10)	7 (21)	0.218
Showed interest in nutrition and healthy eating			
Less than usual	0 (0)	0 (0)	
No more than usual	18 (46)	12 (35)	0.347
More than usual	17 (44)	15 (44)	0.964
Don’t know	0 (0)	0 (0)	
Not applicable	4 (10)	7 (21)	0.218
Chose to eat healthier foods			
Less than usual	1 (3)	0 (0)	0.347
No more than usual	22 (56)	22 (65)	0.470
More than usual	11 (28)	5 (15)	0.164
Don’t know	1 (3)	0 (0)	0.347
Not applicable	4 (10)	7 (21)	0.218
Encouraged parent to buy healthy foods			
Less than usual	0 (0)	1 (3)	0.281
No more than usual	29 (74)	20 (59)	0.159
Less than usual	4 (10)	6 (18)	0.360
More than usual	2 (5)	0 (0)	0.181
Don’t know	2 (5)	1 (3)	0.639
Not applicable	4 (10)	7 (21)	0.218
Frequency study discussed between child and parent			
Every day	5 (13)	3 (9)	0.586
More than twice the past week	10 (26)	8 (24)	0.835
Once or twice this past week	18 (46)	13 (38)	0.495
Not at all this past week	2 (5)	3 (9)	0.533
Not applicable	4 (10)	7 (21)	0.218

Data presented as frequency (percentage); EOS = End-Of-Study. ^a^ As determined by Chi-square.

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
