# Peer review of "The Effectiveness of the Foodbot Factory Mobile Serious Game on Increasing Nutrition Knowledge in Children"

_nutrients, 2020, doi:10.3390/nu12113413_

Round 1

Reviewer 1 Report

This was a study conducted among 39 intervention and 34 control children attending a tech camp to examine the impact of an online nutrition education game (mobile app). Overall this was a well-written manuscript and encouraging findings.  My primary recommendation is to tone down the language in the abstract and discussion given this study was conducted among a very small number of children in a limited age range (with uncertain generalizability to the general population [i.e. kids] in Canada). Specific comments are below.

Abstract

Given that this was only conducted among 39 kids, I think the sentence “Foodbot Factory is an effective and useful learning tool to support nutrition education in children” should be toned down.

Results

LN 283 Did these students who completed the student differ from those who were initially recruited (and the broader population in Canada) in demographics, such as gender, race/ethnicity, SES…

The Discussion needs to be toned down as well to reflect that this was conducted among a small number of children within a limited age range (e.g. the results are not necessarily generalizable to all school-aged children).  This was also conducted at a Tech Camp and therefore it is unclear if results would differ if conducted in a more general population. Limitations should also reflect this.

Tables- Please double check the formatting for tables- the numbers did not always line up with the corresponding variable names.

Table 2.  It would be helpful to know what the max scores are to put these values in context.

Reviewer 2 Report

This is an interesting paper which showed how effective the use of game play can be in nutrition education  especially for children. Generally, the paper was written well with a great detail of methodological framework however I have a number of concerns 1. Table 1 showed that over 60% of the study population have not used nutrition mobile apps and this could have affected the findings of the study- how was this addressed? 2. How many of the subjects were aware of the new Canada's Food Guide and how many have used them prior to the study? 3. What is the relevance of the BMI Percentile categories in this study? The method of assessment should be included in the study. 4. How was the selection and performance bias addressed in the study?
